# Temporal partitioning and spatiotemporal avoidance among large carnivores in a human-impacted African landscape

Charlotte E. Searle[1,2]*, Josephine B. Smit[3,4], Jeremy J. Cusack[5], Paolo Strampelli[1,2], Ana Grau[2], Lameck Mkuburo[3,6], David W. Macdonald[1], Andrew J. Loveridge[1], Amy J. Dickman[1,2]

**1** Wildlife Conservation Research Unit, Department of Zoology, University of Oxford, The Recanati-Kaplan Centre, Tubney, United Kingdom, **2** Ruaha Carnivore Project, Iringa, Tanzania, **3** Southern Tanzania Elephant Program, Iringa, Tanzania, **4** Department of Psychology, University of Stirling, Stirling, United Kingdom, **5** Centro de Modelación y Monitoreo de Ecosistemas, Universidad Mayor, Santiago, Chile, **6** Tanzanian Elephant Foundation, Kilimanjaro, Tanzania

* charlotte.searle@zoo.ox.ac.uk

**Data Availability Statement:** All relevant data are within the manuscript and its Supporting Information files.

## Abstract

Africa is home to some of the world's most functionally diverse guilds of large carnivores. However, they are increasingly under threat from anthropogenic pressures that may exacerbate already intense intra-guild competition. Understanding the coexistence mechanisms employed by these species in human-impacted landscapes could help shed light on some of the more subtle ways in which humans may impact wildlife populations, and inform multi-species conservation planning. We used camera trap data from Tanzania's Ruaha-Rungwa landscape to explore temporal and spatiotemporal associations between members of an intact East African large carnivore guild, and determine how these varied across gradients of anthropogenic impact and protection. All large carnivores except African wild dog (*Lycaon pictus*) exhibited predominantly nocturnal road-travel behaviour. Leopard (*Panthera pardus*) appeared to employ minor temporal avoidance of lion (*Panthera leo*) in all sites except those where human impacts were highest, suggesting that leopard may have been freed up from avoidance of lion in areas where the dominant competitor was less abundant, or that the need for leopard to avoid humans outweighed the need to avoid sympatric competitors. Lion appeared to modify their activity patterns to avoid humans in the most impacted areas. We also found evidence of avoidance and attraction among large carnivores: lion and spotted hyaena (*Crocuta crocuta*) followed leopard; leopard avoided lion; spotted hyaena followed lion; and lion avoided spotted hyaena. Our findings suggest that large carnivores in Ruaha-Rungwa employ fine-scale partitioning mechanisms to facilitate coexistence with both sympatric species and humans, and that growing human pressures may interfere with these behaviours.

**Funding:** Scholarship funding for CS and PS was provided by the University of Oxford NERC Environmental Research DTP (https://www.environmental-research.ox.ac.uk). AD is funded by a Recanati-Kaplan Fellowship (https://www.wildcru.org). Additional funding was awarded to CS for this research from National Geographic Society Early Career Grants (grant number EC-348C-18; https://www.nationalgeographic.org/funding-opportunities/grants/), Cleveland Metroparks Zoo Africa Seed Grants (https://www.clevelandmetroparks.com/zoo), Chicago Zoological Society Chicago Board of Trade (CBOT) Endangered Species Fund (https://www.czs.org/Chicago-Zoological-Society/Conservation-Leadership/CBOT-Endangered-Species-Fund), and Pittsburgh Zoo & PPG Aquarium Conservation & Sustainability Fund (https://www.pittsburghzoo.org/conservation/). The funders had no role in study design, data collection and analysis, decision to publish, or preparation of the manuscript.

**Competing interests:** The authors have declared that no competing interests exist.

## Introduction

Competitive interactions have important fitness consequences for individuals and populations [1], and can trigger cascading impacts across wider ecological systems [2]. Competition can be particularly acute among large carnivores, as they often compete for the same space and resources [3] and have an increased likelihood of lethal interactions due to their predatory adaptations [4]. As a result, interspecific competition can have a strong influence on the distribution, density, and habitat use of large carnivore populations, and thus has important implications for their conservation and management [5].

Intra-guild competition among large carnivores can be indirect, through species needing to access limited shared resources (exploitative competition), or direct, in the form of harassment, kleptoparasitism, or killing (interference competition) [1, 6, 7]. Many large carnivores make use of partitioning mechanisms to minimise the negative consequences of this competition [1]. However, while some species may be able to avoid competitors entirely through spatial segregation, this strategy can bear a high fitness cost by limiting access to vital resources [8]. As such, finer-scale avoidance–such as fine-scale spatial avoidance [9], temporal avoidance [10], or dietary partitioning [11]–can be a valuable strategy for subordinate competitors to facilitate coexistence.

Africa is home to the most functionally diverse guild of large carnivores [12], and its members are therefore exposed to a wide range of potential competitive interactions. Competitive dominance within the African large carnivore guild is primarily determined by body size, with the outcome of individual encounters depending on a number of factors, including group size [13, 14]. Lion (*Panthera leo*) and spotted hyaena (*Crocuta crocuta*) are the dominant carnivores in most African ecosystems, acting as one another's main competitors for resources [15] and exhibiting top-down pressure on their smaller-bodied competitors. Kleptoparasitism and direct killing by lions have been shown to contribute to high mortality rates [16] and population suppression [17] for spotted hyaena. However, competition between the two species is not asymmetrical, with factors including spotted hyaena clan size [17] and individual behavioural differences [18] influencing the outcome of competitive interactions. Lion are also known to attack and kill leopard (*Panthera pardus*; 17), which have been shown to move to denser habitats when in close proximity to lion [19] and actively avoid areas where the probability of encountering the dominant competitor is highest [20]. However, multiple studies have found no evidence of lion suppressing leopard at a population level [20, 21, 22], and it has been theorised that the species may be forced to share space in more anthropogenically impacted areas [23]. Spotted hyaena also typically dominate over leopard [24], and there is evidence of temporal partitioning between the two species [25]. Both lion and spotted hyaena are also known to kill, steal food from, and display aggression towards cheetah (*Acinonyx jubatus*) [26, 27] and African wild dog (*Lycaon pictus*) [28, 29]. Very little is known about intra-guild interactions of striped hyaena (*Hyaena hyaena*), but it is likely that they share a similar competitive status to brown hyaena (*Parahyaena brunnea*), below spotted hyaena and lion [30].

Regardless of their competitive status, all of Africa's large carnivores are increasingly under threat from habitat fragmentation, conflict with humans, and declining prey populations [31]. As a result, members of the African large carnivore guild have been extirpated from large parts of their former range, with range contractions estimated at 94% for lion, 93% for African wild dog, 92% for cheetah, 48–67% for leopard [32], 27% for brown hyaena, 24% for spotted hyaena, and 15% for striped hyaena [33]. In addition to directly impacting survival, anthropogenic pressures are likely to exacerbate already intense intra-guild competition among these species. Habitat fragmentation has been shown to lead to higher levels of spatial overlap among large carnivores [34], limiting options for spatial avoidance. The threat of human-

induced mortality can also trigger large carnivores to partition their activities to avoid direct encounters with humans [35], which may interfere with temporal avoidance of competitors. In addition, declining prey populations may force an increase in dietary overlap among large carnivores [36]–which would increase exploitative competition and potential antagonistic interactions–or force subordinates to shift their diet toward livestock [37], which can intensify conflict with humans. However, relatively little is still known about the different ways in which human disturbance can impact large carnivore interactions, which may hinder effective conservation of these species in areas where they persist among and alongside humans.

Given the ongoing decline of Africa's large carnivores, there is a need to understand how coexistence mechanisms employed by these species vary across modern, human-shaped landscapes, and how they may be affected by anthropogenic impacts. Such knowledge is key to better understand species responses to these novel landscapes, and help inform multi-species and landscape-scale conservation planning. In this study, we used camera trap data from across Tanzania's Ruaha-Rungwa landscape to explore temporal and spatiotemporal associations between members of an intact East African large carnivore guild, including leopard, lion, spotted hyaena, striped hyaena, African wild dog and cheetah. We used activity overlap analysis to explore temporal partitioning among species across habitats, land use types, and gradients of anthropogenic impact, and temporal spacing analysis to investigate spatiotemporal associations between species with sufficient capture data (lion, leopard, and spotted hyaena). By incorporating data from two habitat types within a National Park, a trophy hunting area (Game Reserve), a community-managed area (Wildlife Management Area), and village land, we explore how intra-guild associations between large carnivores vary across a mixed-use, human-impacted area typical of modern African conservation landscapes.

## Materials and methods

### Ethics statement

Data collection consisted of non-invasive camera trapping. Fieldwork was carried out under research permits 2018-368-NA-2018-107, 2018-126-NA-97-20, 2019-96-ER-97-20 and 2019-424-NA-2018-184, granted by the Tanzania Commission for Science and Technology (COST-ECH; Dar es Salaam, Tanzania; rclearance@costech.or.tz) and Tanzania Wildlife Research Institute (TAWIRI; Arusha, Tanzania; researchclearance@tawiri.or.tz).

### Study area

The Ruaha-Rungwa landscape is a mixed-use landscape of over 45,000 km$^2$ located in central-southern Tanzania (Fig 1). The complex is comprised of an unfenced network of protected areas and unprotected land, including Ruaha National Park (NP; allocated for non-consumptive tourism); Rungwa, Kizigo, and Muhesi Game Reserves (GRs; consumptive use through trophy hunting tourism); community-managed MBOMIPA and Waga Wildlife Management Areas (WMAs; consumptive and non-consumptive use); and a number of other less-strictly protected areas (Game Controlled Areas and Open Areas). The landscape is home to a complete East African large carnivore guild [38], including the continent's southernmost population of striped hyaena [39].

### Data collection

**Systematic camera trap surveys.** We used data from camera trap surveys at four sites in the Ruaha-Rungwa landscape: systematic camera trap grids were deployed in an *Acacia-Commiphora* site in Ruaha NP (45 stations, average spacing 1.96 km), in the miombo woodland of

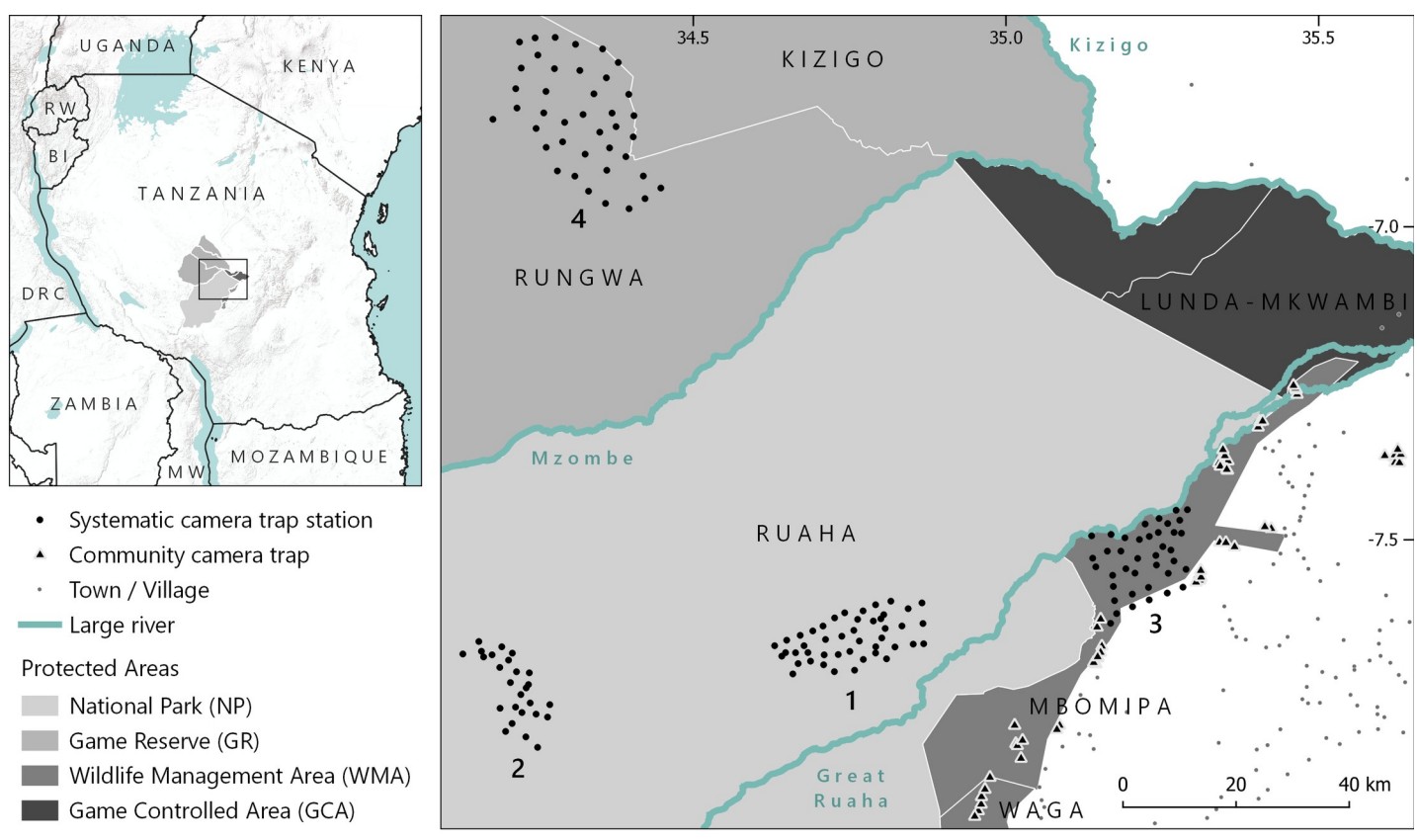

**Fig 1.** Left: the location of the Ruaha-Rungwa landscape within Tanzania (World Terrain Base by Esri, USGS, & NOAA). Right: detail of the four systematic camera trap grids, comprising a total of 151 camera pairs, and the 56 unpaired cameras deployed in village land as part of a community camera-trapping programme (made with Natural Earth). Systematic camera trap grids were located in (1) Ruaha NP *Acacia-Commiphora* habitat, (2) Ruaha NP miombo habitat, (3) MBOMIPA WMA *Acacia-Commiphora* habitat, and (4) Rungwa GR miombo habitat.

western Ruaha NP (26 stations, average spacing 1.88 km), and in an *Acacia-Commiphora* site within MBOMIPA WMA (40 stations, average spacing 2.08 km) during the 2018 dry season; and in the miombo woodland-dominated Ikiri block of Rungwa GR (40 stations, average spacing 3.46 km) in the 2019 dry season (Fig 1). See S1 File and reference [40] for detailed information on survey design and the four survey grids.

The systematic surveys used various models of motion-activated camera (Cuddeback Professional Color Model 1347 & X-Change Color Model 1279, Non Typical Inc., Wisconsin, USA; HC500 HyperFire, Reconyx, Wisconsin, USA). The majority of cameras used xenon white flash to improve the clarity of markings in photos for individual identification. All but one station used paired cameras, to maximise the probability of photographing both flanks of animals passing through the station. Cameras were mounted in protective cases and secured with binding wire to prevent damage and loss to both animals and humans.

We set stations along roads and major game trails or near water points, to maximise captures of large carnivores. Cameras were mounted on trees at a height of 30–40 cm, and deployed for a total of 2–3 months (see S1 File).

Of the four systematic survey sites, the site in MBOMIPA WMA was the most anthropogenically impacted. Although the area was receiving a relatively high level of anti-poaching patrols at the time of data collection, after several years of absence of law enforcement effort [41], it was nevertheless the site most strongly affected by illegal human impacts (including

poaching, charcoal production, and livestock encroachment), as a result of its location directly adjacent to unprotected land and the lack of tourism activities taking place in the WMA at the time of study.

**Community camera traps.** Data from village land were obtained from 56 camera traps deployed non-systematically as part of the Ruaha Carnivore Project's community camera-trapping programme [42]. To facilitate comparisons with the systematic camera trap grids, the data employed were restricted to the 2018 and 2019 dry seasons (June to November inclusive; Fig 1). See S2 File for detailed information on the location and activity of community camera traps.

Community camera traps were deployed on land belonging to 16 villages within the Iringa region. Cameras were placed opportunistically to maximise captures of large carnivores and other wildlife, mostly along trails or water points, and were comprised of both xenon and infra-red flash cameras (Cuddeback Professional Color Model 1347, Non Typical Inc., Wisconsin, USA; HC500 HyperFire, Reconyx, Wisconsin, USA; Scout Guard, HCO Outdoor Products, Georgia, USA).

As sunrise and sunset times do not vary significantly during the dry season in the study area, we did not standardise observations to account for this variation [43].

## Analyses

We used camera trap data to investigate activity patterns (defined as the active period) of the six large carnivore species found in Ruaha-Rungwa: leopard, lion, spotted hyaena, striped hyaena, African wild dog, and cheetah. We assessed activity pattern overlap between these species to identify potential temporal partitioning mechanisms [44, 45], and investigated temporal spacing between detections of different species at stations where they co-occurred, to shed light on spatiotemporal avoidance and attraction among large carnivores in the landscape [46].

As our camera traps were placed predominantly along roads and major trails, it is important to note that the activity we discuss in this study relates specifically to the times at which the study species travel along these habitat features.

**Activity overlap.** We determined activity patterns for each species, and estimated the coefficient of overlap ($\Delta$) between each large carnivore species pair at each site and between different sites for the same species, using the package *overlap* [44] in R version 3.6.2 [47] and RStudio version 1.3.959 [48]; input data can be found in S3 File). Each systematic camera trap grid was treated as a separate site, and all community camera traps in village land were grouped and treated as a single site for this analysis.

To avoid pseudoreplication, captures recorded within 30 minutes of the previous capture of the same species at the same station were excluded from analysis, with the earliest capture retained, unless the animals captured could be confidently distinguished as different individuals [49]. Individuals were identified based on unique coat patterns and other distinguishing features, such as manes, scars, and whisker spots [40, 50]. Where possible, sex was assigned to photographed lion and leopard so that differences in activity patterns of males and females could be investigated for these species.

The overlap estimator for each species pair was selected based on the sample size of the species with the fewest captures at that grid, as recommended based on simulations [45, 51]: the $\Delta_1$ estimator was used if the species with the smallest number of captures at that site had fewer than 75 observations; otherwise, the $\Delta_4$ estimator was used (Table 1).

We estimated confidence intervals for each coefficient of overlap via smoothed bootstrapping with 10,000 resamples [51]. Confidence intervals with an upper limit > 1 were corrected on a logistic scale and back-transformed.

**Table 1. Number of large carnivore capture events at each survey site.**

| Species | Leopard | | | | Lion | | | | Spotted hyaena | Striped hyaena | African wild dog | Cheetah |
|---|---|---|---|---|---|---|---|---|---|---|---|---|
| | M | F | Unk. | All | M | F | Unk. | All | | | | |
| Ruaha NP *Acacia-Commiphora* | 141 | 90 | 22 | 253 | 227 | 239 | 22 | 488 | 1525 | 0 | 71 | 8 |
| Ruaha NP miombo woodland | 24 | 21 | 17 | 62 | 22 | 63 | 7 | 92 | 327 | 0 | 36 | 2 |
| MBOMIPA WMA *Acacia-Commiphora* | 48 | 40 | 12 | 100 | 39 | 43 | 1 | 83 | 427 | 54 | 51 | 6 |
| Rungwa GR miombo woodland | 14 | 17 | 19 | 50 | 60 | 49 | 1 | 110 | 337 | 0 | 5 | 3 |
| Village land (community camera traps) | 34 | 11 | 53 | 98 | 38 | 32 | 17 | 87 | 353 | 48 | 59 | 3 |
| Total | 261 | 179 | 123 | 563 | 386 | 426 | 48 | 860 | 2969 | 102 | 222 | 22 |

For lion and leopard, the number of capture events featuring males (M), females (F), individuals of unknown sex (Unk.), and all captures combined (All) are shown. We define capture events as photographic captures taken at least 30 minutes from the previous capture of that species at that camera (for unpaired cameras) or station (for paired cameras), unless the animal photographed could be confidently distinguished as a new individual.

We used the function *compareCkern* in package *activity* [52] to test whether activity patterns were significantly different for the same species across different sites; for different species at the same site; and between males and females of the same species at the same site (for lion and leopard only).

**Spatiotemporal avoidance and attraction.** We used a temporal spacing analysis developed by Cusack et al. (2017) [46] to investigate whether large carnivores exhibited intra-guild spatiotemporal avoidance or attraction at stations where they co-occurred. Input data consisted of all captures of leopard, lion, and spotted hyaena across all grids combined, as we did not obtain sufficient captures to analyse avoidance and attraction at each site separately (input data can be found in S4 File). Insufficient captures of striped hyaena, African wild dog, and cheetah were obtained for this analysis, and these species were therefore excluded.

For each species pair, we investigated whether one species (species B) was more or less likely than expected to be detected in the 12 hours before and after a capture of the other species (species A), divided into one-hour units.

For all camera stations where the two species co-occurred, we calculated the minimum time before and after each capture of species A to capture species B at the same camera station. Captures of species A were excluded from analysis if they were followed or preceded more closely by another capture of the same species than by a capture of species B. Recorded minimum times were then aggregated into one hour bins, limited to 12 hours before or after the reference capture.

We randomised detection times of species B 1000 times to obtain an expected detection distribution, by selecting a date at random from the survey period of that camera station and sampling a time from the observed activity pattern probability density function for that species. These expected detection probabilities were then compared to the observed detection probabilities via a standard permutation test, to test whether the observed detection probabilities were more or less than expected if the temporal spacing between detections of species A and species B at a camera station was random. We applied a Bonferroni correction to estimated p-values to account for multiple tests.

## Results

### Survey effort

Data from 151 systematically-deployed paired camera stations and 56 non-systematic unpaired camera stations yielded a total of 2,969 unique captures of spotted hyaena, 860 of lion, 563 of leopard, 222 of African wild dog, 102 of striped hyaena, and 22 of cheetah

(Table 1). Leopard, lion, spotted hyaena, African wild dog, and cheetah were captured across all systematic grids and in village land, while striped hyaena were only captured in MBOMIPA WMA and village land [50]. Maps of large carnivore capture events can be found in S5 File.

## Activity overlap

**Intra-specific differences between sites.** All large carnivores in our study area, with the exception of African wild dog, exhibited predominantly nocturnal activity (Fig 2).

Leopard activity in the *Acacia-Commiphora* of Ruaha NP and in Rungwa GR followed a bimodal pattern, with an evening peak around 21:00–22:00 and an early morning peak around 04:00–05:00 (Fig 2A). In the miombo woodland of Ruaha NP the species exhibited only an evening peak, and then a gradual drop off in activity. In the more human-impacted areas (MBOMIPA WMA and village land), the peak of leopard activity was shifted towards the hours after midnight (02:00–03:00). Leopard activity was significantly different (p < .05) between both sites in Ruaha NP and the site in MBOMIPA WMA, and between both sites in Ruaha NP and village land, and highly significantly different (p < .001) between the *Acacia-Commiphora* site of Ruaha NP and village land (Table 2). There was no significant difference between the activity of male and female leopard in any of the study sites–although note that sex identification was limited in village land as a result of camera positioning, making effective investigation of sex differences difficult in this site (Table 3).

Lion were most active in the very early morning (around 02:00–04:00) in all sites except village land, where they exhibited an earlier peak of activity just after midnight (Fig 2B). Lion activity was significantly different in the miombo woodland of Ruaha NP compared to all other sites (Table 2); however, this is likely a result of a pride of three females regularly resting in front of one of the cameras by a water source during the day, which also gave rise to a higher level of daytime activity recorded in this site. Lion activity was also significantly different between MBOMIPA WMA and the *Acacia-Commiphora* area of Ruaha NP, Rungwa GR, and village land, and between Rungwa GR and village land. There was a significant difference in activity patterns of male and female lions in MBOMIPA WMA, where female lions were most active later in the morning than males (Table 3).

Spotted hyaena exhibited a very consistent drop-off and pick-up in activity around sunrise and sunset (07:00 and 19:00), respectively (Fig 2C). Activity was fairly consistent throughout the night-time hours in both sites within Ruaha NP, while in Rungwa GR, MBOMIPA WMA, and village land there was a peak in activity in the early morning, between 03:00 and 05:00, which was most pronounced in village land. Spotted hyaena activity was significantly different between all sites except the two different habitats in Ruaha NP; this difference was highly significant for both habitats in Ruaha NP versus both MBOMIPA WMA and village land, but not Rungwa GR (Table 2).

Striped hyaena exhibited a bimodal activity pattern in village land, with peaks at around 22:00 and 04:00 (Fig 2D). In MBOMIPA WMA, the species exhibited a significantly different activity pattern, with activity instead concentrated around midnight (Table 2).

African wild dog showed a consistent peak in activity around sunrise, although this effect is likely to have been amplified by pack behaviour (Fig 2E). Insufficient captures of African wild dog were obtained in Rungwa GR to assess activity in this site. African wild dog activity patterns were significantly different between all sites where comparisons were possible (Table 2).

For all sites combined, cheetah activity peaked in the evening, at around 22:00 (Fig 2F). There were insufficient captures of cheetah to investigate differences in activity patterns between sites or temporal overlap with other species.

**Inter-specific differences within sites.** Activity patterns of leopard and lion were highly significantly different within both habitat types in Ruaha NP and significantly different in

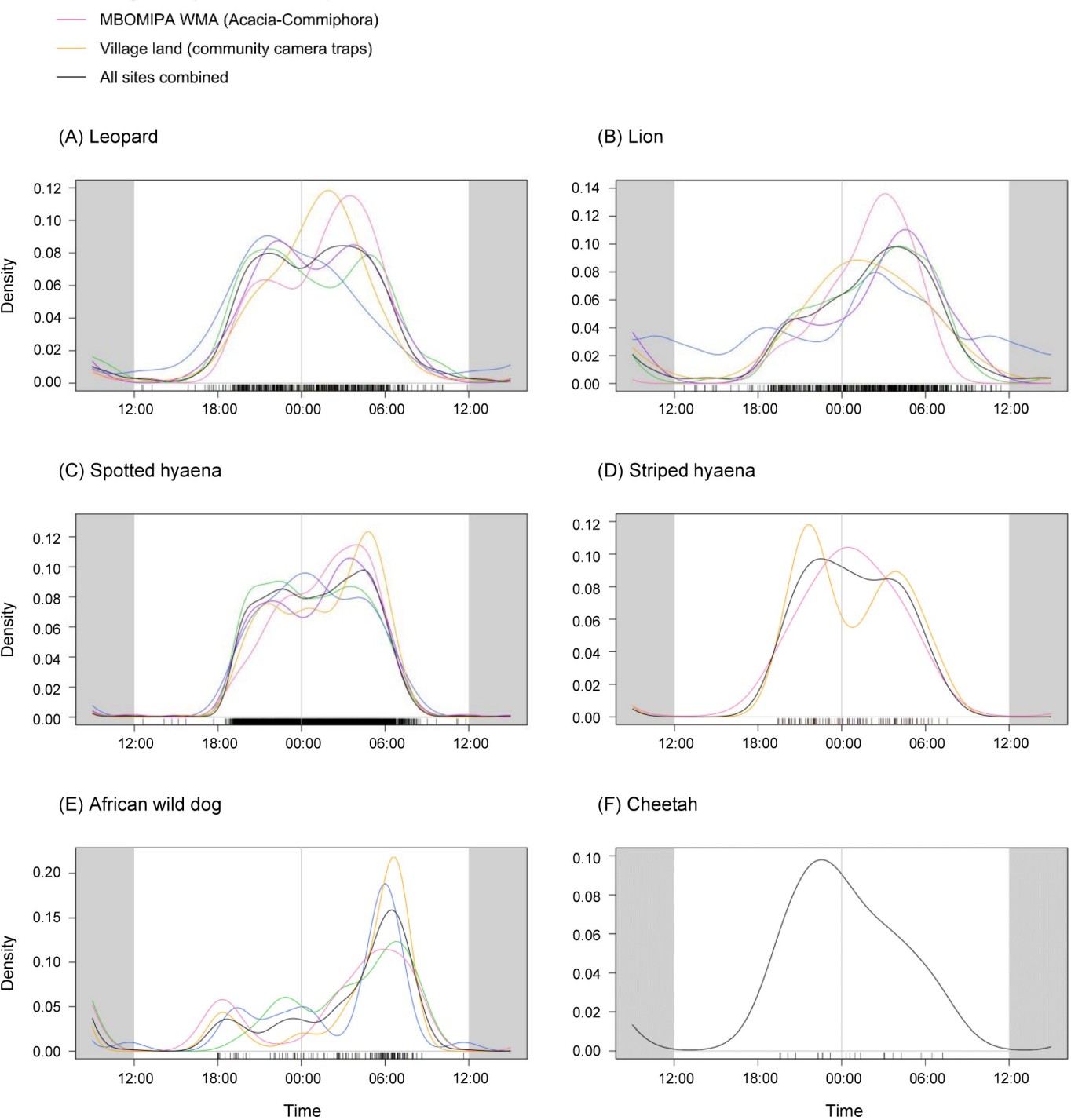

**Fig 2. Kernel density function plots of large carnivore activity in Ruaha-Rungwa, centred on midnight (00:00).** Plotted using package *overlap* [44]. Kernel density functions were fitted to all unique capture events of (A) leopard, (B) lion, (C) spotted hyaena, (D) striped hyaena, (E) African wild dog, and (F) cheetah, using camera trap data from four systematic camera trap grids, and from cameras deployed non-systematically in village land. Striped hyaena were only recorded in MBOMIPA WMA and village land (community camera traps), while insufficient captures of wild dog were obtained in Rungwa GR (n = 5) for activity to be modelled in this site. Due to a low number of overall captures (n = 22), cheetah activity is only plotted for all sites combined.

**Table 2. Coefficients of overlap (Δ) for each large carnivore species between the different survey sites.**

| Site A | Site B | Leopard | Lion | Spotted hyaena | Striped hyaena | African wild dog |
|---|---|---|---|---|---|---|
| Ruaha NP *Acacia-Commiphora* | Ruaha NP miombo woodland | 0.83 (0.74–0.90) | 0.70 (0.61–0.78) | 0.93 (0.89–0.96) | - | 0.63 (0.50–0.74) |
| Ruaha NP *Acacia-Commiphora* | MBOMIPA WMA *Acacia-Commiphora* | 0.81 (0.71–0.89) | 0.79 (0.70–0.87) | 0.86 (0.81–0.90) | - | 0.74 (0.63–0.84) |
| Ruaha NP *Acacia-Commiphora* | Rungwa GR miombo woodland | 0.89 (0.80–0.95) | 0.88 (0.81–0.93) | 0.91 (0.86–0.96) | - | - |
| Ruaha NP *Acacia-Commiphora* | Village land (community camera traps) | 0.75 (0.65–0.83) | 0.85 (0.76–0.92) | 0.87 (0.82–0.91) | - | 0.66 (0.52–0.78) |
| Ruaha NP miombo woodland | MBOMIPA WMA *Acacia-Commiphora* | 0.73 (0.61–0.85) | 0.63 (0.53–0.74) | 0.86 (0.80–0.91) | - | 0.58 (0.44–0.71) |
| Ruaha NP miombo woodland | Rungwa GR miombo woodland | 0.81 (0.68–0.92) | 0.70 (0.60–0.79) | 0.89 (0.84–0.94) | - | - |
| Ruaha NP miombo woodland | Village land (community camera traps) | 0.76 (0.63–0.88) | 0.74 (0.64–0.84) | 0.86 (0.80–0.92) | - | 0.67 (0.54–0.79) |
| MBOMIPA WMA *Acacia-Commiphora* | Rungwa GR miombo woodland | 0.84 (0.72–0.93) | 0.76 (0.65–0.85) | 0.90 (0.85–0.94) | - | - |
| MBOMIPA WMA *Acacia-Commiphora* | Village land (community camera traps) | 0.84 (0.74–0.93) | 0.80 (0.68–0.90) | 0.88 (0.82–0.93) | 0.79 (0.65–0.90) | 0.72 (0.58–0.83) |
| Rungwa GR miombo woodland | Village land (community camera traps) | 0.80 (0.69–0.91) | 0.78 (0.67–0.88) | 0.90 (0.85–0.94) | - | - |

Confidence intervals (shown in parentheses) were estimated via smoothed bootstrapping with 10,000 resamples; confidence intervals with an upper limit > 1 were corrected on a logistic scale and back-transformed. Overlap analysis was only carried out for survey site interactions where the sample size of the site with fewest captures was > 30 (see Table 1). Cells highlighted in light green indicate significant differences (p < .05) in activity between the two sites, while cells highlighted in dark green indicate highly significant differences (p < .001).

Rungwa GR–where lion consistently exhibited more morning activity than leopard, and leopard more evening activity than lion–but did not differ significantly in MBOMIPA WMA or village land (Table 3).

Leopard and spotted hyaena exhibited significantly different activity patterns in both Ruaha NP sites, where spotted hyaena were more active than leopard in the early hours of the morning, and were more strictly nocturnal. They also exhibited highly significantly different activity in village land, where spotted hyaena were more crepuscular than leopard (Table 3).

Spotted hyaena appeared to be more active in the evenings than lion, and activity patterns of the two species were significantly different everywhere except MBOMIPA WMA (this difference was highly significant in all sites except village land; Table 3).

Striped hyaena activity was significantly different from lion across both grids where striped hyaena were recorded (MBOMIPA WMA and village land; Table 3), while leopard and striped hyaena activity were significantly different in village land (Table 3). Striped hyaena and spotted hyaena activity did not differ significantly (Table 3).

African wild dog exhibited significantly different activity patterns from all other large carnivores for all sites where comparisons were possible (Table 3).

Plots of within- and between-species activity overlap can be found in S6 File.

## Spatiotemporal avoidance and attraction

Leopard were significantly more likely to be captured than expected in the hour before a lion capture at the same station (Fig 3A), and lion were more likely to be captured than expected in the hour after a leopard capture (Fig 3B; although the latter result was not significant). A similar pattern was also found when detection probabilities were estimated for units of 30 minutes,

**Table 3. Coefficients of overlap (Δ) for all large carnivore species pairs at each survey site.**

| Survey location | Leopard Lion | Leopard Spotted hyaena | Leopard Striped hyaena | Leopard African wild dog | Lion Spotted hyaena | Lion Striped hyaena |
|---|---|---|---|---|---|---|
| Ruaha NP *Acacia-Commiphora* | 0.82 (0.75–0.88) | 0.89 (0.84–0.93) | - | 0.69 (0.58–0.78) | 0.82 (0.77–0.86) | - |
| Ruaha NP miombo woodland | 0.68 (0.56–0.79) | 0.81 (0.70–0.90) | - | 0.53 (0.36–0.69) | 0.66 (0.58–0.74) | - |
| MBOMIPA WMA *Acacia-Commiphora* | 0.86 (0.76–0.94) | 0.91 (0.84–0.97) | 0.81 (0.68–0.91) | 0.57 (0.44–0.69) | 0.91 (0.83–0.97) | 0.77 (0.63–0.89) |
| Rungwa GR miombo woodland | 0.78 (0.66–0.90) | 0.88 (0.79–0.95) | - | - | 0.81 (0.72–0.88) | - |
| Village land (community camera traps) | 0.85 (0.75–0.94) | 0.78 (0.69–0.86) | 0.74 (0.61–0.86) | 0.39 (0.28–0.51) | 0.81 (0.73–0.88) | 0.77 (0.64–0.88) |
| Survey location | Lion African wild dog | Spotted hyaena Striped hyaena | Spotted hyaena African wild dog | Striped hyaena African wild dog | Leopard (M) Leopard (F) | Lion (M) Lion (F) |
| Ruaha NP Acacia-Commiphora | 0.78 (0.67–0.87) | - | 0.66 (0.56–0.76) | - | 0.82 (0.72–0.91) | 0.90 (0.85–0.95) |
| Ruaha NP miombo woodland | 0.55 (0.40–0.69) | - | 0.57 (0.43–0.72) | - | 0.87 (0.69–0.99) | 0.63 (0.47–0.78) |
| MBOMIPA WMA Acacia-Commiphora | 0.55 (0.42–0.68) | 0.84 (0.73–0.93) | 0.55 (0.43–0.66) | 0.48 (0.35–0.61) | 0.87 (0.75–0.97) | 0.79 (0.63–0.92) |
| Rungwa GR miombo woodland | - | - | - | - | - | 0.83 (0.70–0.93) |
| Village land (community camera traps) | 0.48 (0.36–0.61) | 0.83 (0.72–0.92) | 0.49 (0.37–0.59) | 0.44 (0.30–0.57) | - | 0.77 (0.62–0.90) |

Confidence intervals (shown in parentheses) were estimated via smoothed bootstrapping with 10,000 resamples; confidence intervals with an upper limit > 1 were corrected on a logistic scale and back-transformed. Overlap analysis was only carried out for species interactions where the sample size of the species with fewest captures was > 30 (see Table 1). Cells highlighted in light green indicate significant differences (p < .05) in activity between the two species, while cells highlighted in dark green indicate highly significant differences (p < .001).

instead of one hour (S7A Fig in S7 File). These results suggest that lion may be following leopard, or that leopard may be vacating areas due to lion moving in.

We also found evidence of leopard avoiding lion, as lion were significantly less likely to be captured than expected in the hour before a leopard capture (Fig 3B). Although not significant, leopard were also less likely to be captured than expected in the hour after a lion capture (Fig 3A).

We found evidence of lion avoiding spotted hyaena, with lion being significantly less likely to be captured than expected on nights when spotted hyaena were captured (Fig 3D). Spotted hyaena were also more likely to be captured than expected two or three hours after a lion capture (Fig 3C; although not significantly), suggesting that they may be following lion.

Finally, our results suggest that spotted hyaena may be following leopard, as leopard were significantly more likely to be captured than expected in the hour before a spotted hyaena capture (Fig 3E; a pattern also found with a time unit of 30 minutes; see S7A Fig in S7 File). Although we found no evidence of leopard avoiding hyaena at the one hour time unit, spotted hyaena were significantly less likely to be captured than expected in the six hours before a leopard capture (see S7C Fig in S7 File).

## Discussion

We employed camera trap data to reveal insights into interactive processes between large carnivores in the mixed-use Ruaha-Rungwa conservation landscape in Tanzania. We compared activity patterns–specifically, movement along roads and trails–across different habitats and

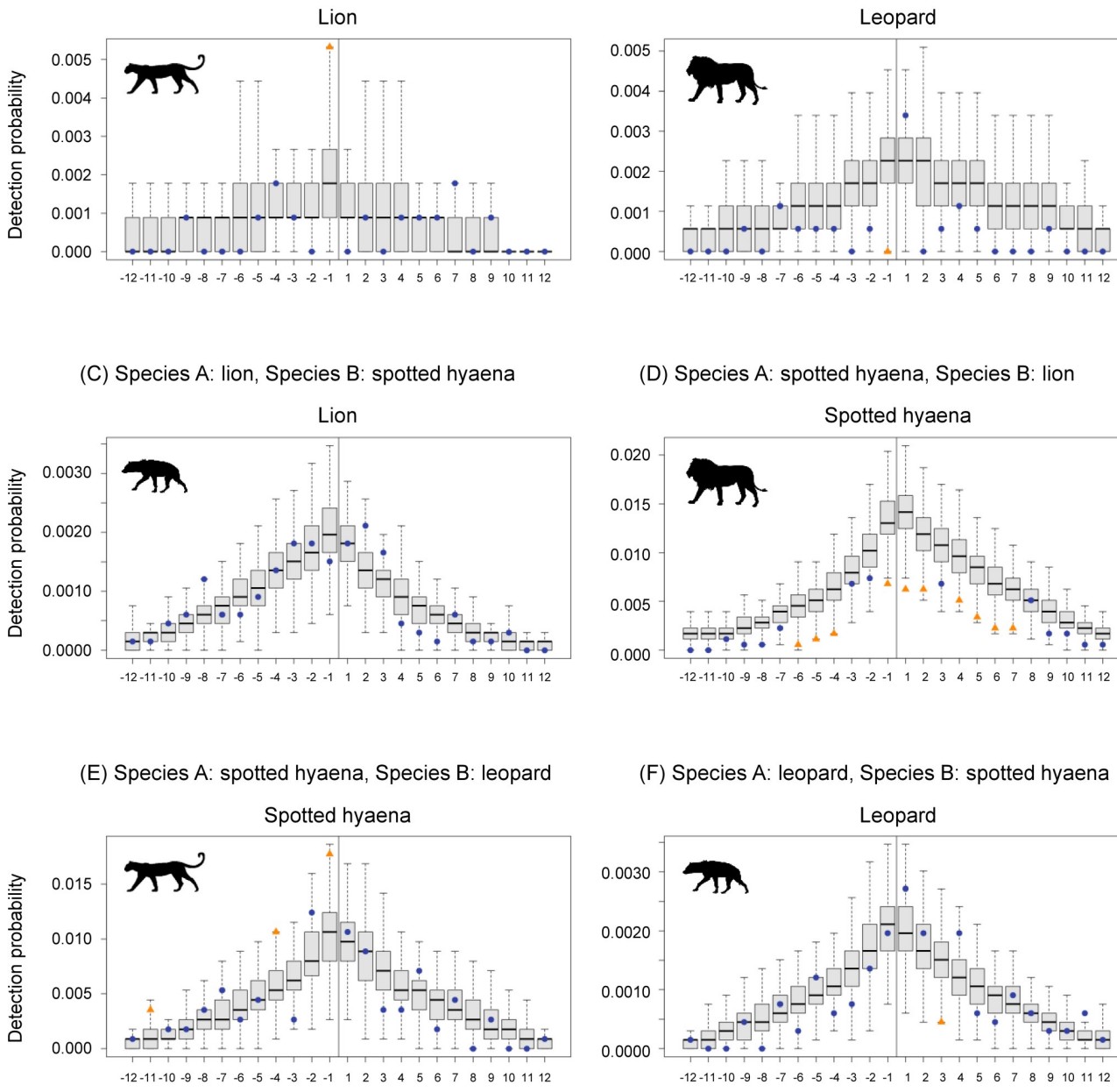

**Fig 3. Detection probability of species B (shown via silhouettes) in the 12 hours before and after a capture of species A (at hour 0) at the same station, divided into one-hour units.** Boxplots show the expected probability of detecting species B in each hour before and after capture of species A if there were no relationship between detections of species A and species B, while points indicate the observed detection probability of species B for each hour before and after capture of species A. Expected detection probabilities were derived by randomly sampling 1000 times from the observed activity pattern probability density function for that species. Observed detection probabilities that differ significantly (p < 0.05) from the expected probability of detection are shown with orange triangles; those that do not differ significantly are shown with blue dots.

land use types, and found evidence of temporal partitioning by large carnivores with intra-guild competitors. We also found evidence of significant avoidance and attraction behaviours between leopard, lion, and spotted hyaena.

## Activity patterns and species overlap

We found no evidence of sex differences in leopard activity patterns, in contrast to a study in Tanzania's Udzungwa Mountains [25]. We did, however, find evidence of intra-specific activity difference among leopard across study sites, with the species' peak of activity occurring deeper into night-time hours in the more human-impacted sites.

Leopard and lion activity differed significantly in Ruaha NP and Rungwa GR. Supported by our findings that lion followed leopard and leopard avoided lion, this suggests that leopard may employ minor temporal partitioning to avoid the dominant competitor in this landscape. However, the shift of leopard activity towards the early morning in more human-impacted sites resulted in a greater activity overlap with lion. This may be a result of the need to avoid humans outweighing the need to avoid dominant competitors in areas where human activity is higher, as suggested elsewhere [23]. Leopard have been shown to partition temporally from humans [53, 54] and avoid bushmeat poachers [55], and 47% of humans inside the WMA illegally that were captured by the systematic camera traps were photographed between the hours of 18:00 and 00:00, versus 13% between 00:00 and 06:00 (this study). A similar trade-off has been observed in African wild dog, where temporal avoidance of humans left packs more vulnerable to competition from lion and spotted hyaena [56].

However, this pattern may also, at least in part, reflect leopard being freed up from avoidance of the dominant competitor in areas with fewer lions. Although lion population density was higher in MBOMIPA WMA ($4.06 \pm 1.03$ per 100 km$^2$) than the miombo woodland of Rungwa GR ($2.25 \pm 0.52$ per 100 km$^2$) and Ruaha NP ($1.75 \pm 0.62$ per 100 km$^2$; see S8 File), there appears to be a strong density gradient within the WMA, with 93% of lion detections in this site occurring within 5 km of the Great Ruaha River, further from the boundary with unprotected village land (see S5.1B Fig in S5 File). It is also highly probable that lion persist at very low densities in unprotected village land, as a result of widespread habitat conversion and degradation, low prey availability, and high anthropogenic mortality [57]. In these areas, lion densities may therefore be sufficiently low that leopard no longer need to adjust their activity patterns to avoid harmful interactions with lion–particularly as this would increase their exposure to humans–and can instead rely more heavily on reactive avoidance strategies [9].

The significant difference between leopard and spotted hyaena activity in the site with highest spotted hyaena density (Ruaha NP *Acacia-Commiphora*: $10.8 \pm 1.08$ per 100 km$^2$; see S8 File) may indicate some temporal partitioning by leopard to avoid competitive interactions with spotted hyaena, as observed elsewhere in Tanzania [25]. This is supported by our finding that spotted hyaena appear to follow leopard where the two species co-occur, a pattern which has also been observed in South Africa [58].

Most lion activity was concentrated in the early hours of the morning except in village land, where they exhibited an earlier peak of activity around midnight. This predominantly nocturnal activity is consistent with numerous studies of the species elsewhere in Africa [10]. The activity shift observed in village land likely reflects avoidance of humans–either through lion modifying their overall activity patterns, or reducing their use of roads at times when humans are more likely to be encountered along them. Such avoidance of human-dominated areas by lion during the day has also been observed in Kenya [35]. Although we did not find evidence of partitioning with humans by lion in the WMA grid, we would expect any human avoidance to be less noticeable in this site than in village land, both because the majority of lions in this grid were recorded in the best-protected area along the Great Ruaha River (see S5.1B Fig in S5 File), and because illegal human incursions in the WMA took place mostly before midnight, and so did not coincide with the consistent peak of lion activity we observed within protected areas.

Spotted hyaena activity was significantly different in Ruaha NP versus the more human-impacted sites of MBOMIPA WMA and village land, where the species exhibited a more pronounced peak in activity in the hours before sunrise. This may be a result of individuals in unprotected or more encroached areas having to travel extensively along roads–and thus pass in front of the camera traps–to return to safe areas where they can hide during the day before sunrise, when human activity increases substantially. Similar avoidance of humans has been observed among spotted hyaena in livestock grazing areas, where they have been shown to shift toward more strictly nocturnal activity [59] and travel faster and farther [60]. However, the differences in activity across sites may also reflect differences in prey activity patterns across sites, with spotted hyaena modifying their activity to better access prey.

Although we obtained a low number of detections of cheetah, our data suggest that the species may exhibit predominantly nocturnal activity in Ruaha-Rungwa. This contrasts with a number of studies that have found the species to be largely diurnal or crepuscular [10, 61, 62]. However, research in Namibia has found cheetah to show increased nocturnal activity as ambient temperatures increase [63], and a study in Botswana's Okavango Delta revealed an unexpectedly high rate of nocturnal activity for the species, with higher rates of night-time activity on nights with more moonlight [64]. Although we did not account for moon phase in this study, our findings nonetheless add to a growing body of evidence that cheetah may be more nocturnal than previously thought.

For all species, it is important to note that the apparent differences in activity revealed by our camera trap data may in fact reflect differences in the times at which species travel along roads between the study sites, rather than differences in overall activity. Species may still be active when off-road–for example, wild dog in northern Botswana have been shown to select roads when travelling, but ignore roads when running at high speeds or resting [65]. However, this may nevertheless reflect behavioural adaptations to avoid competition, as species may shift their movement away from roads to avoid intra-guild competitors or humans. This is a particular possibility for cheetah, as there is evidence that the species can detect lion presence and adjust their behaviour to avoid lion spatially, but may not need to do so in areas with denser vegetation [66].

## Spatiotemporal avoidance and attraction

The results of our temporal spacing analysis suggest that leopard employ fine-scale avoidance of lion in areas where they co-occur in Ruaha-Rungwa. Lion are known to attack and kill leopard [20], steal kills from the smaller-bodied felid [67], and impact cub survival and recruitment rates in leopard populations [68]. However, multiple studies have found leopard to not spatially segregate from lion over larger areas [20, 21, 22]. In our study landscape, this is likely a result of the species' shared habitat preferences [69]: large-scale avoidance of lion would prevent leopard from accessing many vital resources, particularly in human-impacted areas where resources are more limited, and this may pose a fitness cost to leopard that outweighs the benefits of complete avoidance of the dominant competitor [23]. Instead, by prioritising resource acquisition and responding reactively to the presence of lion through fine-scale behavioural adaptations, leopard are able to minimise the risk of antagonistic encounters while maintaining access to preferred habitats [66].

In addition to the fine-scale temporal avoidance observed here, a number of additional coexistence mechanisms have been observed in other leopard populations, including fine-scale spatial avoidance in the presence of lion [9, 19], avoiding areas where the probability of encountering lion is highest [20], minor temporal partitioning [21], dietary niche differentiation [70], and prey caching to avoid kleptoparasitism [67, 71]. Thus, although this study

provides evidence of fine-scale spatiotemporal avoidance and possible minor temporal partitioning, it is likely that leopard in Ruaha-Rungwa also make use of other adaptations that could not be identified by this study.

We also found evidence that spotted hyaena follow leopard in Ruaha-Rungwa. This mirrors the results of a similar study in South Africa [58], and is likely to be a behaviour employed by spotted hyaena to increase kleptoparasitism and scavenging opportunities. Kleptoparasitism by spotted hyaena is often mainly targeted at leopard [9], and previous research has found spotted hyaena to be responsible for 50% of kleptoparasitism suffered by the species [67]. High rates of kleptoparasitism are associated with reduced reproductive success among female leopards [67]; the loss of prey to competitors, and particularly to spotted hyaena, therefore represents a threat to leopard fitness. Nevertheless, as spotted hyaena are ubiquitous across the Ruaha-Rungwa landscape, it is likely to be impossible for leopard to fully spatially segregate from them at coarse scales, even in unprotected areas [58]. As a result, behavioural adaptations like those employed to avoid lion may be necessary for leopard to also avoid competitive interactions with spotted hyaena, and thus minimise the fitness consequences of coexistence.

Our findings suggest that spotted hyaena also actively follow lion, and that lion avoid spotted hyaena in Ruaha-Rungwa. This could reflect spotted hyaena being more reliant on scavenging than active predation in the study area–a shift that has been observed in areas of high lion density [72]–and following lion to maximise their chances of being able to take over a kill once it has been abandoned [73]. The corresponding avoidance shown by lion may suggest attempts by lion to minimise these antagonistic interactions. However, this apparent avoidance could also be reflective of periods of high spotted hyaena activity along roads coinciding with lion activity being focused off-road, which may occur when lion are on a kill.

Interestingly, we did not find evidence of reciprocal following behaviour by lion toward spotted hyaena, despite both species being known to steal kills from one another [74]. Lion and spotted hyaena have been found to actively track one another in Serengeti NP [62], and in a previous temporal spacing study in the *Acacia-Commiphora* area of Ruaha NP [46]. However, the previous study in Ruaha NP only found evidence of reciprocal following between the two species in the first of the two dry seasons studied. Although is important to note that the inclusion of data from other parts of the Ruaha-Rungwa landscape in this study may have obscured any reciprocal following within individual sites, we obtained more than 50% of both lion and spotted hyaena captures in the *Acacia-Commiphora* of Ruaha NP (Table 1); as such, it is likely that we would have picked up on any evidence of lion following spotted hyaena in this site if this were still taking place. Comparing our findings to this previous research therefore illustrates the variability of intra-guild interactions, and suggests that there may have been a shift from reciprocal to one-sided following behaviour between lion and spotted hyaena between 2013 and 2018. A change of this kind could be indicative of an increased ratio of spotted hyaena to lion in Ruaha-Rungwa–as a result of an increase in spotted hyaena population or a declining lion population, or a combination of the two–or of the sex ratio of lion in the study area becoming more female-weighted, both of which confer a higher competitive status to spotted hyaena [13, 17]. Evidence of lion ceasing to follow spotted hyaena may also point to changes in diet, as mutual attraction between the species can indicate mutual attraction to shared prey [62]. Such changes may be linked to the ongoing drying of the once-perennial Great Ruaha River due to upstream water mismanagement, which has resulted in a considerable loss of dry season habitat [75, 76], and warrant further investigation into the impacts of this ecological shift on Ruaha-Rungwa's wildlife populations.

It should be noted that our inferences about the direction of intra-guild interactions were informed by existing knowledge of associations between different large carnivore species [46],

and we acknowledge that camera trap data can only provide limited insights into complex competitive interactions–particularly as these are not limited to roads, and often take place over very short timescales. In Ruaha-Rungwa, movement data from GPS or VHF collars would be a valuable tool to investigate the fine-scale coexistence mechanisms being used by large carnivores [9], and provide a clearer picture of how individuals may be modifying their movements to avoid humans [35, 53, 54].

### Conservation implications

Despite the evidence we found of spatiotemporal avoidance among members of Ruaha-Rungwa's large carnivore guild, population densities of leopard, lion, and spotted hyaena were nonetheless highly correlated across our systematic survey sites (leopard-lion: 0.97; leopard-hyaena: 0.94; lion-hyaena: 0.88; see S8 File; 38). Our results therefore highlight the importance of the observed fine-scale avoidance mechanisms employed by large carnivores to minimise the negative impacts of intra-guild competition, and secure access to limited shared resources, as well as to avoid competition with humans [77].

However, these adaptations are likely to become both increasingly important and more difficult for species to employ as the availability of intact habitat and prey continue to decline as a result of anthropogenic pressures [33, 78], and large carnivores are forced into smaller, more fragmented areas [38]. In this context, information on how species modify their behaviour to co-exist with one another and with humans, as provided in this study, can inform conservation management plans in increasingly human-modified landscapes, and help secure Africa's large carnivore diversity. In particular, efforts should be made to prioritise multi-species conservation efforts; maintain heterogeneity in prey species, habitats, and habitat features; and minimise activities which would interfere with known coexistence mechanisms during critical periods of activity for species.

## Supporting information

**S1 File. Survey grid summary information.**
(PDF)

**S2 File. Location and activity of community camera traps.**
(PDF)

**S3 File. Activity analysis input data.**
(CSV)

**S4 File. Temporal spacing input data.**
(CSV)

**S5 File. Maps of large carnivore capture events.**
(PDF)

**S6 File. Plots of within- and between-species activity overlap.**
(PDF)

**S7 File. Full temporal spacing analysis results.**
(PDF)

**S8 File. Ruaha-Rungwa large carnivore population density.**
(PDF)

## Acknowledgments

Fieldwork for this research was carried out under permits 2018-368-NA-2018-107, 2018-126-NA-97-20, 2019-96-ER-97-20 and 2019-424-NA-2018-184, granted by the Tanzania Commission for Science and Technology (COSTECH) and Tanzania Wildlife Research Institute (TAWIRI). We would like to thank the Government of Tanzania, TAWIRI, Tanzania National Parks Authority (TANAPA), Tanzania Wildlife Management Authority (TAWA), and Idodi-Pawaga MBOMIPA WMA for their support of this research. We also thank the field staff of the Southern Tanzania Elephant Program (STEP) and Ruaha Carnivore Project (RCP) for their assistance with data collection, and MBOMIPA WMA Village Game Scouts, TANAPA Rangers, TAWA Game Scouts, and Tanzania Big Game Safaris for their assistance during fieldwork.

## Author Contributions

**Conceptualization:** Charlotte E. Searle, Paolo Strampelli.

**Data curation:** Charlotte E. Searle, Josephine B. Smit, Ana Grau.

**Formal analysis:** Charlotte E. Searle.

**Funding acquisition:** Charlotte E. Searle, Amy J. Dickman.

**Investigation:** Charlotte E. Searle, Josephine B. Smit, Paolo Strampelli, Ana Grau, Lameck Mkuburo.

**Methodology:** Charlotte E. Searle, Jeremy J. Cusack, Paolo Strampelli.

**Software:** Charlotte E. Searle, Jeremy J. Cusack.

**Supervision:** David W. Macdonald, Andrew J. Loveridge, Amy J. Dickman.

**Validation:** Charlotte E. Searle, Jeremy J. Cusack.

**Writing – original draft:** Charlotte E. Searle.

**Writing – review & editing:** Charlotte E. Searle, Josephine B. Smit, Jeremy J. Cusack, Paolo Strampelli, Ana Grau, Lameck Mkuburo, David W. Macdonald, Andrew J. Loveridge, Amy J. Dickman.

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
