## [Decision Letter · Decision Letter 0]

15 Jun 2021

PONE-D-21-15323

Temporal partitioning and spatiotemporal avoidance among large carnivores in a human-impacted African landscape

PLOS ONE

Dear Dr. Searle,

Thank you for submitting your manuscript to PLOS ONE. After careful consideration, we feel that it has merit but does not fully meet PLOS ONE’s publication criteria as it currently stands. Therefore, we invite you to submit a revised version of the manuscript that addresses the points raised during the review process.

Pay attention to reviewers comments, especially to reviewer #2, who highlighted some issues in the manuscript. Check the emphasis you put on the literature of competitive interactions on large carnivores and less emphasis on how human disturbance impact these competitive relationships. The introduction need references to support some statements. There are some confusing parts that need to be clarified.

We look forward to receiving your revised manuscript.

Kind regards,

Paulo Corti, Ph.D.

Academic Editor

PLOS ONE

Journal Requirements:

2. We note that Figures 1 and S5 in your submission contain map images which may be copyrighted. All PLOS content is published under the Creative Commons Attribution License (CC BY 4.0), which means that the manuscript, images, and Supporting Information files will be freely available online, and any third party is permitted to access, download, copy, distribute, and use these materials in any way, even commercially, with proper attribution. For these reasons, we cannot publish previously copyrighted maps or satellite images created using proprietary data, such as Google software (Google Maps, Street View, and Earth). For more information, see our copyright guidelines: http://journals.plos.org/plosone/s/licenses-and-copyright.

a) You may seek permission from the original copyright holder of Figures 1 and S5 to publish the content specifically under the CC BY 4.0 license.  

Reviewers' comments:

Reviewer #1: This study uses camera trap data to examine the effects of human impact on temporal and spatiotemporal associations between some of Tanzania’s large carnivores. The authors found evidence of avoidance and attraction between the various carnivores, and that both leopard and lion altered these behaviours in the areas with high human impact. This suggests that, at least for road-travelling behaviours, human activity may interfere with spatiotemporal partitioning mechanisms that carnivore guilds use to facilitate coexistence with one another.

This is a very well written manuscript that clearly describes its methodology and findings. The authors are also transparent about the limitations of their data and the implications of their results. I enjoyed reading this paper, and I only have one very minor suggestion which the authors can take or leave as desired. I recommend this paper be accepted for publication.

Multi-panelled figures 2 & 3: the axes titles and labels for these figures is a bit small and hard to read. I suggest increasing the font size here. To help avoid the figures becoming overly crowded, I suggest that instead of having each figure panel’s axes titled individually, instead have a larger axes title placed in a central position between the panels (since in each figure, the x and y axis for the panels is the same).

Reviewer #2: Increase in anthropogenic impacts have been resulted in disruption of ecological process across the world and competitive interactions among large carnivores are one. Present study able to depicts how human disturbance can disrupt inter-predator relationship. Following are the comments from me:

Introduction of the manuscript is robust but have put to much emphasis on the literature of competitive interactions on large carnivores and less emphasis on how human disturbance impact these competitive relationships except one paragraph. Also, introduction lack References at certain statement. For example, Line 48, 49, 50 involves description of competition lack any reference to support the statement. Another statement in line 72 and 73 i.e., it has been theorised that the species may be forced to share space in more anthropogenically impacted areas. Does this statement can be applied in protected area which lack much human disturbance and leopards rely much on cover rather than human impacted area?

Under data collection heading paragraph includes line number from 136-142 is confusing. For example, in line number 136 i.e., We used data from camera trap surveys at four sites in the Ruaha-Rungwa landscape (38), what does 38 means is not clear. In Analysis section from line number 227 to 234 not a single reference has been provided to support that this analysis has been used in some earlier study. In survey effort a total independent capture of the species under study has been provided since study studies at different sites hence independent captures at different sites should also mentioned in survey effort. Also, how much captures of male and female leopards were got has not mentioned in survey effort which also need to be stated.

Authors have able to justified their findings thoroughly in discussion. But discussion lack light on spatio-temporal attraction in relation to different habitat which authors have sampled in RNP. Do these habitats differ in their structure and visibility which could further affect the outcome of the competitive interactions?

In nutshell present study is well written and have robust statical analysis and need minor revision.

---

## [Author Response · Author response to Decision Letter 0]

25 Jul 2021

Please see uploaded file "Large carnivore interactions_Response to reviewers" for our response to the editor and reviewers' comments.

---

## [Decision Letter · Decision Letter 1]

18 Aug 2021

Temporal partitioning and spatiotemporal avoidance among large carnivores in a human-impacted African landscape

PONE-D-21-15323R1

Dear Dr. Searle,

We’re pleased to inform you that your manuscript has been judged scientifically suitable for publication and will be formally accepted for publication once it meets all outstanding technical requirements.

Kind regards,

Paulo Corti, Ph.D.

Academic Editor

PLOS ONE

**Comments to the Author**

Reviewer #2: Present manuscript have rigorously answered the questions pertaining to large carnivore interactions in human dominated landscape. Authors have incorporated all the comments raised by me and manuscript can be accepted for publication in present form.

---

## [Editor Report · Acceptance letter]

31 Aug 2021

PONE-D-21-15323R1 

Temporal partitioning and spatiotemporal avoidance among large carnivores in a human-impacted African landscape 

Dear Dr. Searle:

I'm pleased to inform you that your manuscript has been deemed suitable for publication in PLOS ONE. Congratulations! Your manuscript is now with our production department. 

Kind regards, 

on behalf of

Dr. Paulo Corti 

Academic Editor

PLOS ONE